# BRENA: Brain-inspired Hierarchical Neural Alignment Framework for Visual Decoding from EEG Signals

## Abstract

Decoding human visual experiences from neural signals is crucial for understanding the relationship between brain activity and perceptual representations, driving the advancement of brain-computer interface (BCI) applications. Existing visual decoding methods typically adopt a global alignment paradigm for brain-visual alignment, which may not account for the intricate functional specialization of the human visual cortex, where distinct cortical areas are selectively sensitive to different visual information. In this work, we propose **BRENA**, a **BR**ain-inspired hi**E**rarchical **N**eural **A**lignment framework by simultaneously aligning both region-level and global brain representations with visual embeddings for robust and accurate brain decoding. Unlike prior approaches that rely purely on global pooled representations, BRENA proposes an Adaptive Local Neural Alignment Module to explore fine-grained correspondence between brain channel features and visual semantic units, allowing for better exploitation of brain signals by modeling region-specific feature selectivity. Additionally, a set of perceptual weights are adaptively generated to guide more target-aware alignment. We further integrate a global neural alignment module to achieve hierarchical brain-visual alignment, capturing complementary region-level and global neural patterns. Experiments demonstrate that BRENA not only outperforms existing methods across subjects and settings but also reveals region-level brain selectivity for visual stimuli by establishing meaningful local mappings between neural channels and diverse visual sub-patterns.

## 1 Introduction

Decoding visual content from brain activity aims to infer the content or properties of visual stimuli from neural recordings (Georgopoulos et al., 1986; Kay et al., 2008; Guo et al., 2025; Wu et al., 2025), which underpins the design of brain–computer interfaces and driving progress in cognitive neuroscience (Nicolas-Alonso & Gomez-Gil, 2012; Kriegeskorte & Douglas, 2018). Stimulus-evoked neural activity is mainly recorded by EEG, MEG, and fMRI. Among these non-invasive neural recording modalities, EEG has attracted broad attention owing to its portability and millisecond temporal resolution (Schirrmeister et al., 2017; Song et al., 2022; 2023).

Recently, various EEG-based visual decoding methods have been proposed, which aim to use deep learning models to decode the corresponding visual stimuli perceived during the recording process. These methods typically align EEG embeddings with image representation space to achieve visual retrieval or reconstruction (Benchetrit et al., 2023; Scotti et al., 2024; Zhang et al., 2025). However, existing methods typically use a global alignment paradigm for brain-visual alignment, which may not account for the human visual cortex's functional specialization and distributed organization.

Previous studies (Roe et al., 2012; Cant & Goodale, 2007; Coggan et al., 2017) on human perception mechanisms have revealed that *distinct visual cortical regions are selectively sensitive to different visual patterns*. As shown in Figure 1, the two-streams hypothesis in neuroscience divides the visual system into a ventral "what" pathway and a dorsal "where" pathway (Ungerleider, 1982). The ventral pathway, driven predominantly by parvocellular input and coursing along the ventral visual cortex, selectively encodes fine detail for object recognition (Merigan & Maunsell, 1993; Nassi &

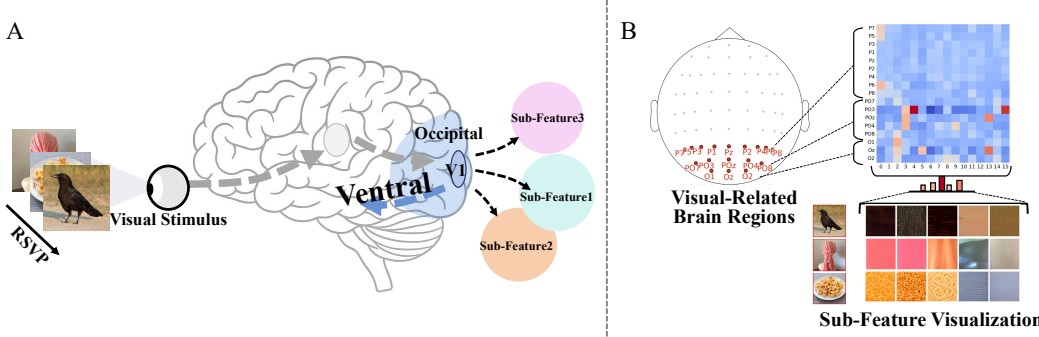

Figure 1: (A) Overview of information flow during Rapid Serial Visual Presentation (RSVP) and the hierarchical processing mechanism of multi-level visual information along the ventral stream. (B) Visualization of brain's region-specific selectivity in visual decoding. The heatmap matrix depicts the mapping from EEG channels (cortical regions) to different target visual sub-features, and larger values indicate greater sensitivity of that channel to the corresponding sub-feature.

Callaway, 2009). Along the ventral stream, low-level features are parsed in specialized early (V1/V2 (Bridge et al., 2005)) and mid-level (V4 (Hamker, 2005)) subregions, and their outputs are progressively integrated across hierarchical stages (Lerner et al., 2001; DiCarlo et al., 2012; Roe et al., 2012). Despite the brain's hierarchical visual perceptual process, existing decoding methods compress multi-regional neural signals into a single global representation and conduct global alignment with compressed visual semantics, underrepresenting how different parts of the visual cortex work together. This loss of **region-specific selectivity** is particularly problematic for EEG signals that are notorious for low signal-to-noise ratio, hindering the accurate interpretation of neural patterns. Therefore, a significant challenge is to develop a robust visual decoding method that could reflect the brain's inherent hierarchical and cooperative visual processing.

Motivated by the above insights, we propose a **BR**ain-inspired hi**E**rarchical **N**eural **A**lignment (**BRENA**) framework by performing brain-visual alignment at both region-level and global-level for robust brain decoding. Inspired by region-specific feature selectivity of visual cortex, we propose an Adaptive Local Neural Alignment module, which aligns features of different EEG spatial channels to visual sub-features with diverse patterns. To accomplish this, we decompose images into several task-relevant sub-features via a visual resampler, forming semantic subspaces that align more effectively with brain regions. Additionally, a set of adaptive perceptual weights are generated to prioritize sub-features that are most predictive of targets, guiding the local alignment process to focus on discriminative cues. Consequently, BRENA achieves a finer-grained alignment by mimicking the visual cortex's functional specialization. This enables task-specific EEG channels to be activated with corresponding low- and mid-level visual semantics (such as color, texture, and foreground–background spatial relation), while reducing interference from irrelevant channels. In parallel, a global neural alignment module enforces sample-level semantic coherence between images and brain signals via a global contrastive loss. By integrating both local and global neural alignment, BRENA achieves a hierarchical brain-visual alignment with both fine-grained regional neural patterns and global semantics captured, allowing for a richer exploitation of neural data.

Our contributions could be summarized in threefold:

- We propose a brain-inspired hierarchical neural alignment framework by taking inspiration from the brain's functional specialization, performing brain-visual alignment at both region-level and global-level for robust visual decoding.

- An Adaptive Local Neural Alignment Module is proposed to achieve fine-grained brain-visual alignment between EEG channel embeddings and diverse visual sub-features, allowing for better exploitation of brain signals by modeling region-specific feature selectivity.

- Extensive experiments in both intra-subject and inter-subject settings demonstrate the effectiveness of BRENA in visual decoding, outperforming state-of-the-art methods by 7.3% in average top-1 accuracy on image retrieval tasks.

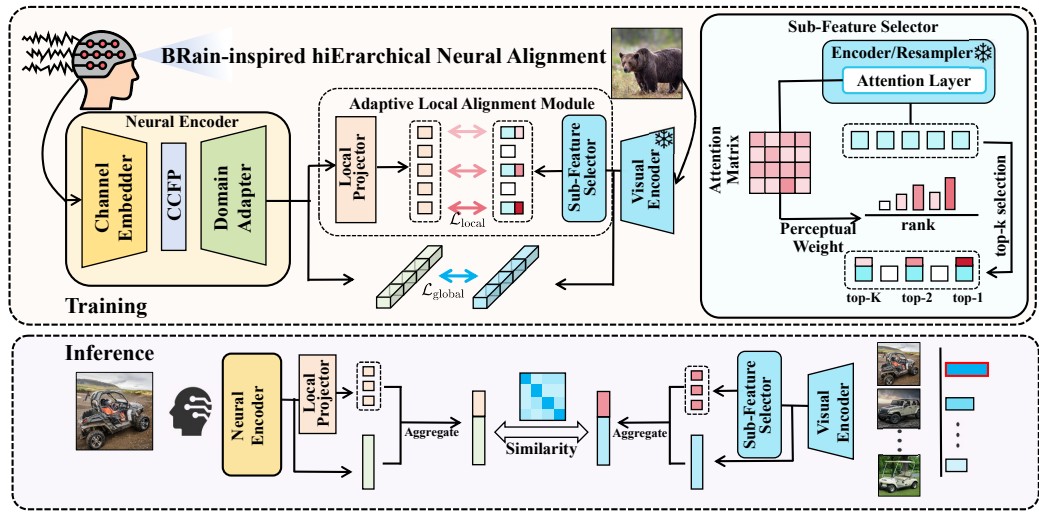

Figure 2: Overview of **BRENA**. An Adaptive Local Neural Alignment Module is introduced for fine-grained alignment between EEG channel embeddings and selected visual sub-features. In addition, a sub-feature selector is adopted to adaptively select the discriminative sub-features guided by adaptive perceptual weights. In parallel, the global alignment at the sample-level is achieved via a global contrastive loss. In the inference stage, the local key sub-features and global features are aggregated to perform retrieval task.

- We conduct brain functional visualization and discover that BRENA reveals region-level brain selectivity through meaningful local mappings between brain channels and diverse visual patterns, grounding the alignment framework in human perceptual mechanisms.

## 2 METHOD

### 2.1 OVERVIEW

Inspired by the visual cortex's region-specific selectivity, we propose a hierarchical neural alignment framework to mimic its functional specialization, activating task-relevant EEG channels with low- and mid-level visual semantics (e.g., color, texture, spatial relations) while suppressing irrelevant signals. As depicted in Figure 2, the framework consists of two primary components: a **Neural Encoder** designed to produce robust EEG representations and an **Adaptive Local Alignment Module** that aligns these region-level representations with visual sub-features, guided by a set of adaptively generated perceptual weights.

Overall, given EEG data $\mathbf{S} \in \mathbb{R}^{C \times T}$ with $C$ channels and $T$ time points and an image $\mathbf{X}^{(i)}$ as the visual stimuli, a trainable Neural Encoder $f_\theta$ is utilized to extract the brain semantic representation $\hat{\mathbf{V}} = f_\theta(\mathbf{S}) = [\hat{\mathbf{v}}_1, \hat{\mathbf{v}}_2, \ldots, \hat{\mathbf{v}}_C] \in \mathbb{R}^{C \times m}$. To obtain the target visual features, we use a frozen pretrained vision encoder $\text{VE}(\cdot)$ to produce decomposed visual sub-features from the image $\mathbf{X}$ as $\mathbf{V} = \text{VE}(\mathbf{x}) = [\mathbf{v}_1, \mathbf{v}_2, \ldots, \mathbf{v}_K] \in \mathbb{R}^{K \times n}$. Our goal is to reduce the modality discrepancy between the brain representations and the corresponding visual sub-features, thereby achieving effective visual decoding.

### 2.2 NEURAL ENCODER

Unlike prior global alignment approaches (Li et al., 2024a) that apply early global mixture across EEG channels, we first perform per-channel feature extraction without cross-channel mixing to preserve region-specific information. Subsequently, these channel-wise neural signals are aggregated to be adaptively mapped to their most relevant sub-visual features. Overall, our EEG encoder consists of three parts: Channel Embedder, Cross-Channel Fusion Projector, and Domain Adapter.

**Channel Embedder.** Previous work (Wang et al., 2025; Li et al., 2024b) demonstrates that frequency-domain features capture both category- and state-dependent information. To leverage this property, we introduce a dual-branch channel encoder that captures joint time–frequency features for enhanced EEG representation. Specifically, for each channel signal segment $\mathbf{s}_j \in \mathbb{R}^T, j \in [1, 2, \cdots, C]$, the frequency branch computes a power–spectrum vector $\mathbf{p}_j = \left| \mathcal{F}\{\mathbf{s}_j\} \right|^2$ by the discrete Fourier transform $\mathcal{F}$, where $|\cdot|^2$ is the element-wise power. Then, we project it to a $m$-dimensional embedding $\mathbf{e}_j^f = \varphi_f(\mathbf{p}_j)$. In parallel, we further extract EEG's temporal characteristic by $\mathbf{e}_j^t = \varphi_t(\mathbf{s}_j)$. In practice, we set $m$ equal to $T$. Then we fuse the two paths by residual addition:

$$\mathbf{z}_j = \mathbf{e}_j^t + \mathbf{e}_j^f + \mathbf{s}_j. \tag{1}$$

**Cross-Channel Fusion Projector (CCFP).** Motivated by the brain's regional specialization for processing low-level visual attributes, we adaptively aggregate functionally related channels using a cross-channel projector. This crucial step enables their precise alignment with subsequent low- and mid-level visual information. Specifically, we concatenate per-channel embeddings and map it to $K$ vectors in the feature space of vision yield $\hat{\mathbf{Z}} = \psi(\mathrm{f}(\mathbf{Z})) \in \mathbb{R}^{K \times n}$, which is termed as the brain semantic embeddings. For simplicity, we utilize a high-capacity linear projection, which could learn data-driven channel combinations that emphasize brain regions most predictive of each target feature. As shown in Figure 1, by visualizing the weight matrix of the fusion mapping layer, we can identify EEG channels that contribute most to specific sub-visual features and compare the relative strength of channel–feature couplings across brain regions.

**Domain Adapter.** To reduce the modality gap between neural features and the target space, we employ a domain adapter on the brain semantic embeddings $\hat{\mathbf{Z}}$. We apply a shared adapter to transform each brain semantic embedding, bridging the representational gap between the brain embeddings and the visual domain: $\hat{\mathbf{v}}_j = \hat{\mathbf{z}}_j + g_\theta(\hat{\mathbf{z}}_j), j \in [1, 2, \ldots, K]$, where the adapter $g_\theta$ is implemented as a lightweight MLP layer.

## 2.3 ADAPTIVE LOCAL ALIGNMENT MODULE

Motivated by the region-specific feature selectivity of the visual cortex, we introduce an Adaptive Local Alignment module to align brain semantic embeddings with diverse visual sub-features. Specifically, we design a sub-feature selector that learns perceptual weights to rank visual sub-features, and introduce a local alignment loss to achieve fine-grained alignment while emphasizing discriminative patterns of the perceived target.

**Visual Sub-Feature Learning.** To mimic the human perception process, we first decompose images into a set of visual sub-features before aligning them with the EEG channel features. Specifically, for a given input image $x^{(i)}$, we use a pretrained visual encoder (Ye et al., 2023) equipped with a visual resampler. The resampler employs several learnable queries to fuse patch embeddings via cross-attention, thereby obtaining a set of visual sub-features $\{\mathbf{v}_i\}_{i=1}^N$.

Subsequently, a sub-feature selector uses perceptual weights to identify the most discriminative sub-features, guiding the local alignment process to focus on the most critical patterns. The perceptual weights could quantify each sub-feature's contribution to the global representation, generating from the attention matrix $\mathbf{A} \in \mathbb{R}^{N \times N}$ from the last cross-attention layer. For the $i$-th sub-feature, we compute its perception weight as the mean attention received across all heads and queries by $w_i = \frac{1}{H} \sum_{h=1}^H \frac{1}{N} \sum_{j=1}^N A_{j,i}^{(h)}$, where $H$ is the number of attention heads. We then select top-$k$ sub-features contributing most to the global representation:

$$\mathcal{I}_{\text{top-}k} = \mathrm{TopK}\left(\{w_i\}_{i=1}^N, k\right). \tag{2}$$

**Local Alignment.** Given the brain semantic embeddings $\{\hat{\mathbf{v}}_i\}_{i=1}^N$ from neural encoder, a local projector $\mathcal{M}(\cdot)$ is designed to obtain brain region representations as:

$$\mathbf{u}_i = \mathcal{M}(\hat{\mathbf{v}}_i), \quad i = 1, \ldots, N. \tag{3}$$

Then, we design a local alignment loss to ensure robust fine-grained alignment. Specifically, for the selected top-$k$ indices $\mathcal{I}_{\text{top-}k}$, we extract the predicted brain-decoded sub-features $\{\mathbf{u}_i\}_{i \in \mathcal{I}_{\text{top-}k}}$

and the corresponding ground-truth image sub-features $\{\mathbf{v}_i\}_{i \in \mathcal{I}_{\text{top-}k}}$ at the same positions. The local alignment loss $\mathcal{L}_{\text{local}}$ guided by perceptual weights could be formulated as:

$$\mathcal{L}_{\text{local}} = -\frac{1}{k} \sum_{i \in \mathcal{I}_{\text{top-}k}} w_i \cdot \frac{\mathbf{u}_i \cdot \mathbf{v}_i}{\|\mathbf{u}_i\|_2 \|\mathbf{v}_i\|_2}. \tag{4}$$

By enforcing local alignment loss $\mathcal{L}_{\text{local}}$, BRENA could activate task-specific EEG channels with corresponding low- and mid-level visual semantics while suppressing interference from irrelevant channels.

## 2.4 TRAINING OBJECTIVES

To preserve sample-level semantics and complement local alignment, we design a **global alignment loss** that aligns the global EEG representations with holistic visual semantics. Given the feature sets $\hat{\mathbf{V}}, \mathbf{V} \in \mathbb{R}^{K \times n}$, we flatten them into global embeddings $\hat{\mathbf{G}}, \mathbf{G}$. We then apply a CLIP-style symmetric contrastive loss on a batch of size $B$. Let the similarity be $s_{ij} = \langle \hat{\mathbf{G}}, \mathbf{G} \rangle / \tau$ with temperature $\tau > 0$. Formally,

$$\mathcal{L}_{\text{global}} = \frac{1}{2} \left[ -\frac{1}{B} \sum_{i=1}^{B} \log \frac{\exp(s_{ii})}{\sum_{j=1}^{B} \exp(s_{ij})} - \frac{1}{B} \sum_{i=1}^{B} \log \frac{\exp(s_{ii})}{\sum_{j=1}^{B} \exp(s_{ji})} \right]. \tag{5}$$

The final training objective $\mathcal{L}_{\text{overall}}$ combines the local and global contrastive loss to enforce cross-modal consistency, yielding discriminative EEG representations: $\mathcal{L}_{\text{overall}} = \mathcal{L}_{\text{global}} + \lambda \mathcal{L}_{\text{local}}$, where $\lambda$ is the trade-off parameter.

# 3 EXPERIMENT

## 3.1 IMPLEMENTATION DETAILS

**Dataset.** We employed the THINGS-EEG dataset (Gifford et al., 2022), a large-scale benchmark for visual neural decoding. It contains EEG recordings from ten participants who viewed object images from the THINGS database under a rapid serial visual presentation (RSVP) paradigm, where each image was displayed for 100 ms followed by a 100 ms blank interval. The training set comprises 1,654 concepts with multiple repetitions per subject, while the test set includes 200 novel concepts with extensive repetitions to ensure high signal-to-noise ratio. We followed the protocol in NICE (Song et al., 2023) to preprocess EEG data.

**Compared Methods.** We compare our proposed BRENA with several state-of-the-art baselines, including BraVL (Du et al., 2023), NICE and its variants (NICE-SA, NICE-GA) (Song et al., 2023), ATM-S (Li et al., 2024a), VE-SDN (Chen et al., 2024), UBP (Wu et al., 2025). We adopt their official implementations and follow the same evaluation protocols to ensure a fair comparison.

**Visual Encoder.** We use the fused sub-features produced by **IP-Adapter**Ye et al. (2023) as the default alignment targets. It applies a lightweight Perceiver Resampler to fuse the ViT-H/14 patch tokens into 16 semantically enriched visual sub-features. More implementation details are shown in Appendix C.

## 3.2 OVERALL PERFORMANCE

Table 1 shows the overall comparison of Top-1 and Top-5 accuracy (%) for 200-way zero-shot retrieval on THINGS-EEG across different baselines. The experiments are conducted in both **intra-subject** setting where training and testing are performed on the same subject, and **inter-subject setting** where models are trained on all but one subject and evaluated on the held-out participant. Overall, BRENA achieves the best intra-subject performance and robust inter-subject generalization.

Table 1: Top-1 and Top-5 accuracy (%) for 200-way zero-shot retrieval on THINGS-EEG. Improvements are reported relative to UBP, where positive values are marked in green (+) and negative values in red (−).

| Method | Subject 1 top-1 | top-5 | Subject 2 top-1 | top-5 | Subject 3 top-1 | top-5 | Subject 4 top-1 | top-5 | Subject 5 top-1 | top-5 | Subject 6 top-1 | top-5 | Subject 7 top-1 | top-5 | Subject 8 top-1 | top-5 | Subject 9 top-1 | top-5 | Subject 10 top-1 | top-5 | Avg top-1 | top-5 |
|---|---|---|---|---|---|---|---|---|---|---|---|---|---|---|---|---|---|---|---|---|---|---|
| **Intra-subject**: train and test on one subject | | | | | | | | | | | | | | | | | | | | | | |
| BraVL | 6.1 | 17.9 | 4.9 | 14.9 | 5.6 | 17.4 | 5.0 | 15.1 | 4.0 | 13.4 | 6.0 | 18.2 | 6.5 | 20.4 | 8.8 | 23.7 | 4.3 | 14.0 | 7.0 | 19.7 | 5.8 | 17.5 |
| NICE | 13.2 | 39.5 | 13.5 | 40.3 | 14.5 | 42.7 | 20.6 | 52.7 | 10.1 | 31.5 | 16.5 | 44.0 | 17.0 | 42.1 | 22.9 | 56.1 | 15.4 | 41.6 | 17.4 | 45.8 | 16.1 | 43.6 |
| NICE-SA | 13.3 | 40.2 | 12.1 | 36.1 | 15.3 | 39.6 | 15.9 | 49.0 | 9.8 | 34.4 | 14.2 | 42.4 | 17.9 | 43.6 | 18.2 | 50.2 | 14.4 | 38.7 | 16.0 | 42.8 | 14.7 | 41.7 |
| NICE-GA | 15.2 | 40.1 | 13.9 | 40.1 | 14.7 | 42.7 | 17.6 | 48.9 | 9.0 | 29.7 | 16.4 | 44.4 | 14.9 | 43.1 | 20.3 | 52.1 | 14.1 | 39.7 | 19.6 | 46.7 | 15.6 | 42.8 |
| ATM-S | 25.6 | 60.4 | 22.0 | 54.5 | 25.0 | 62.4 | 31.4 | 60.9 | 12.9 | 43.0 | 21.3 | 51.1 | 30.5 | 61.5 | 38.8 | 72.0 | 34.4 | 51.5 | 29.1 | 63.5 | 28.5 | 60.4 |
| VE-SDN | 32.6 | 63.7 | 34.4 | 69.9 | 38.7 | 73.5 | 39.8 | 72.0 | 29.4 | 58.6 | 34.5 | 68.8 | 34.5 | 68.3 | 49.3 | 79.8 | 39.0 | 69.6 | 39.8 | 75.3 | 37.2 | 69.9 |
| UBP | 41.2 | 70.5 | 51.2 | 80.9 | 51.2 | 82.0 | 51.1 | 76.9 | 42.2 | 72.8 | 57.5 | 83.5 | 49.0 | 79.9 | 58.6 | 85.8 | 45.1 | 76.2 | 61.5 | 88.2 | 50.9 | 79.7 |
| **Ours** | **61.0** | **87.0** | **56.0** | **81.5** | **56.5** | **87.5** | **57.0** | **84.5** | **46.5** | **74.5** | **56.5** | **87.0** | **58.5** | **86.0** | **75.0** | **91.5** | **50.0** | **80.0** | **65.0** | **92.0** | **58.2** | **85.1** |
| *Improve* | +19.8 | +16.5 | +4.8 | +0.6 | +5.3 | +5.5 | +5.9 | +7.6 | +4.3 | +1.7 | −1.0 | +3.5 | +9.5 | +6.1 | +16.4 | +5.7 | +4.9 | +3.8 | +3.5 | +3.8 | +7.3 | +5.4 |
| **Inter-subject**: leave one subject out for test | | | | | | | | | | | | | | | | | | | | | | |
| BraVL | 2.3 | 8.0 | 1.5 | 6.3 | 1.4 | 5.9 | 1.7 | 6.7 | 1.5 | 5.6 | 1.8 | 7.2 | 2.1 | 8.1 | 2.2 | 7.6 | 1.6 | 6.4 | 2.3 | 8.5 | 1.8 | 7.0 |
| NICE | 7.6 | 22.8 | 5.9 | 20.5 | 6.0 | 22.3 | 6.3 | 20.7 | 4.4 | 18.3 | 5.6 | 22.2 | 5.6 | 19.7 | 6.3 | 22.0 | 5.7 | 17.6 | 8.4 | 28.3 | 6.2 | 21.4 |
| NICE-SA | 7.0 | 22.6 | 6.6 | 23.2 | 7.5 | 23.7 | 5.4 | 21.4 | 6.6 | 22.2 | 7.5 | 22.5 | 3.8 | 19.1 | 8.5 | 24.4 | 7.4 | 22.3 | 9.8 | 29.6 | 7.0 | 23.1 |
| NICE-GA | 5.9 | 21.4 | 6.4 | 22.7 | 5.5 | 20.1 | 6.1 | 21.0 | 4.7 | 19.5 | 6.2 | 22.5 | 5.9 | 19.1 | 7.3 | 25.3 | 4.8 | 18.3 | 6.2 | 26.3 | 5.9 | 21.6 |
| ATM-S | 10.5 | 26.8 | 7.1 | 24.8 | 11.9 | 33.8 | 14.7 | 39.4 | 7.0 | 23.9 | 11.1 | 35.8 | 16.1 | 43.5 | 15.0 | 40.3 | 4.9 | 22.7 | 20.5 | 46.5 | 11.8 | 33.7 |
| UBP | 11.5 | 29.7 | 15.5 | 40.0 | 9.8 | 27.0 | 13.0 | 32.3 | 8.8 | 33.8 | 11.7 | 31.0 | 10.2 | 23.8 | 12.2 | 32.2 | 15.5 | 40.5 | 16.0 | 43.5 | 12.4 | 33.4 |
| **Ours** | **15.0** | **37.5** | **18.5** | **47.0** | **11.0** | **31.5** | **18.5** | **39.5** | **13.5** | **35.0** | **9.5** | **27.5** | **13.5** | **36.0** | **15.0** | **40.0** | **18.0** | **42.0** | **24.5** | **50.0** | **15.7** | **38.7** |
| *Improve* | +3.5 | +7.8 | +3.0 | +7.0 | +1.2 | +4.5 | +5.5 | +7.2 | +4.7 | +1.2 | −2.2 | −3.5 | +3.3 | +12.2 | +2.8 | +7.8 | +2.5 | +1.5 | +8.5 | +6.5 | +3.3 | +5.3 |

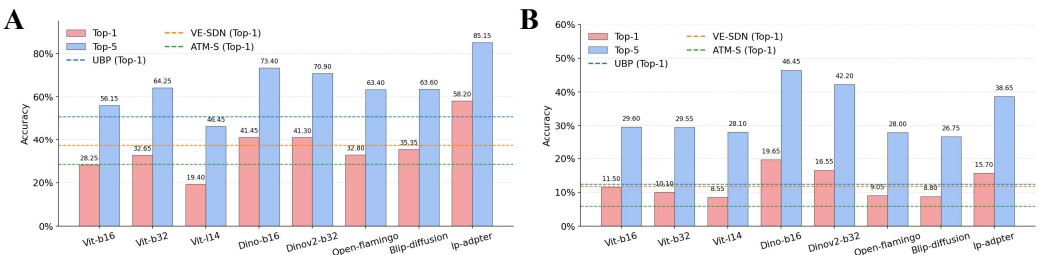

Figure 3: Comparison of Top-1 and Top-5 retrieval accuracies across different pretrained visual targets under 200-way zero-shot setting. Dashed lines indicate Top-1 accuracies from baseline models (UBP, ATM-S, VE-SDN). (A) Intra-subject retrieval accuracy across visual targets. (B) Inter-subject retrieval accuracy across visual targets.

**Intra-subject Performance.** Our method achieves the best performance, with Top-1 and Top-5 accuracies of **58.2%** and **85.1%**, respectively. Compared with UBP (50.9% / 79.7%), this yields absolute improvements of 7.3 and 5.4 points. Gains are consistent across participants: BRENA ranks first or ties for first in the majority of cases, reflecting robustness to within-subject variability. These results suggest that BRENA better captures task-relevant brain-visual correspondences than global alignment paradigm baselines.

**Inter-subject Performance.** Cross-subject generalization is highly challenging due to large individual variability in EEG responses. Even under this distribution shift, our approach achieves the best average Top-5 accuracy of **38.7%**, an absolute gain of 5.3 points over the strongest prior baseline (UBP: 33.4%), and Top-1 accuracy reaches **15.7%**. These findings indicate that BRENA learns shared structure across subjects more effectively.

## 3.3 EFFECT OF VISUAL BACKBONE CHOICE ON EEG-VISION ALIGNMENT

**Visual Encoder.** Figure 3 compares the performance of using visual targets generated from eight visual backbones under the 200-way zero-shot retrieval setting, including ViT-B/16, ViT-B/32, ViT-L/14, DINO ViT-B/16, DINOv2 ViT-B/32, OpenFlamingo, BLIP-Diffusion, and IP-Adapter.

**Intra-subject Performance.** Visual features from IP-Adapter achieved the highest scores (Top-1: 58.2%, Top-5: 85.1%), substantially surpassing all baselines. Self-supervised representations also perform strongly: DINO-B/16 (41.5% / 73.4%) and DINOv2-B/32 (41.3% / 70.9%) clearly outper-

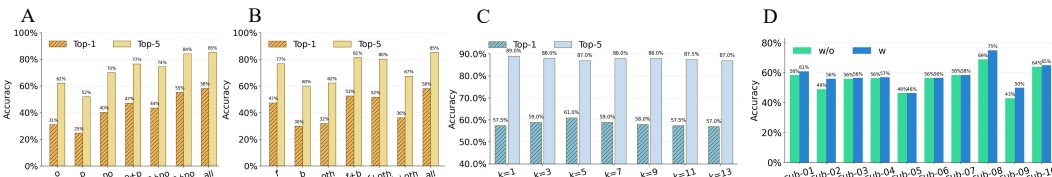

Figure 4: Ablation study. We probe the performance impact of using different brain channel groups (occipital (o), parieto–occipital (po), and parietal (p)) and visual sub-features (foreground (f), background (b), or other (oth) sub-features) in (A) and (B). We also present the impact of the number of sub-features and adaptive local alignment module in (C) and (D). Detailed results are in Appendix.

form CLIP ViT-B models of similar scale. By contrast, CLIP ViT-L/14, despite its larger backbone, underperforms (19.4% / 46.5%), suggesting that excessively subdivided features misalign with the brain's region-specific perceptual pattern. DINO and DINOv2, owing to its stronger representational capacity, performance second only to IP-Adapter and even surpass the second-best baseline. Overall, most BRENA variants with different visual backbones perform well and exceed ATM-S, demonstrating the effectiveness of our sub-level alignment design.

**Inter-subject Performance.** In the inter-subject setting, the ranking remains consistent, with DINO-B/16 achieving the highest performance (19.1% / 46.5%), followed by IP-Adapter (15.7% / 38.7%). Other variants cluster around 8–12% Top-1 and 28–31% Top-5. Notably, the DINO family even surpasses the resampler-based IP-Adapter representations. Although IP-Adapter produces compact, semantically concentrated sub-features by fusing ViT-L/14 tokens, these results suggest the critical of expressiveness of the underlying visual backbone critical.

### 3.4 ABLATION STUDY OF MODEL COMPONENTS

**Ablation of Adaptive Local Alignment Module.** As shown in Figure 4 (D), the utilization of *Adaptive Local Alignment* consistently improves Top-1 accuracy across all ten subjects, with the most notable gains on `sub-02`, `sub-08`, and `sub-09`. The results underscore the importance of regionally-aware local alignment for brain decoding. This approach establishes a fine-grained correspondence to better leverage region-specific neural patterns, thereby providing benefits that are complementary to a global objective.

**Impact of the Number of Visual Sub-features.** We further conduct sensitivity analysis to the number of visual sub-features $k$ on `sub-01` (Figure 4 (C)). The model performance achieves the best accuracy at $k=7$ and then displays a plateau trend. We attribute this to smaller $k$ values having limited capacity to capture discriminative cues, while overly large $k$ dilutes salient information. The stability observed around $k \in [3, 7]$ suggests that the module is robust to the specific choice of $k$, and thus we adopt $k=7$ as a robust default.

**Impact of Brain Channel and Visual Sub-feature Combinations.** We probe how brain channel groups and visual sub-features affect EEG decoding by grouping channels into occipital (o), parieto–occipital (po), and parietal (p), and aligning to foreground (f), background (b), or other (oth) sub-features. As shown in Figure 4 (A), the *po* channels achieve the best performance (40% / 75%). Considering group combinations, the *o* channels contribute the most, and the *o+po* pairing yields the strongest results (55% / 84% ). From Figure 4 (B), we can see that aligning foreground features yields more discriminative representations that benefit retrieval, whereas aligning background features has the opposite effect. Performance improves with increased aligned sub-features.

### 3.5 BRAIN FUNCTIONAL VISUALIZATION

**Temporal Analysis of Neural Activity.** We visualize the temporal dynamics of brain-visual alignment. Specifically, we analyze the mapping matrix learned by CCFP, which relates EEG representations to brain semantic embeddings corresponding to visual sub-features, over consecutive 100 ms windows. As shown in Figure 5, we present two types of visualizations. Figure 5(A) displays the

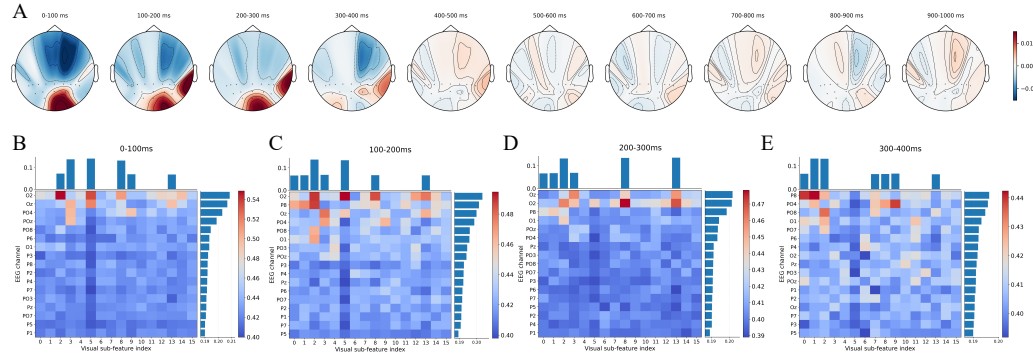

Figure 5: (A) EEG scalp topographies at different time steps (0–1000 ms, 100 ms steps) of sub-01. (B)-(E) Cross-channel fusion projection heatmaps and channel–feature associations, where rows correspond to EEG channels, columns to visual sub-feature indices with flanking bar plots indicating row/column marginals. Each row (EEG channel) is row-normalized to highlight selectivity across sub-features, and rows are re-ordered by their per-row significance.

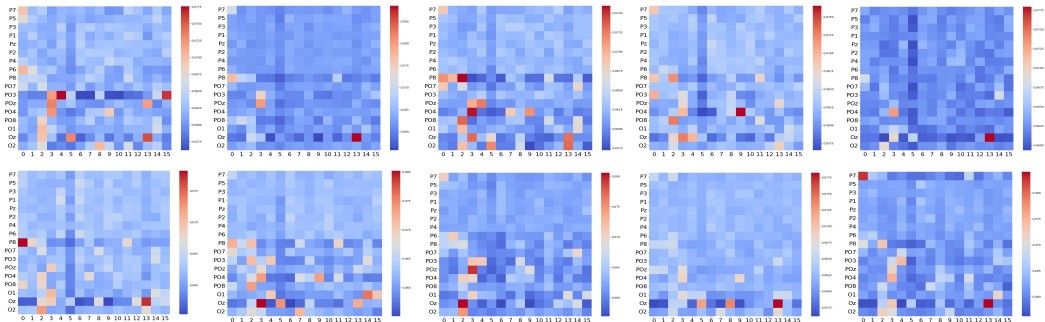

Figure 6: Cross-channel fusion projection heatmaps of 10 subjects. Panels are ordered from top-left to bottom-right as sub-(1–10).

average mapping values for each channel. We selectively show the first four matrices in Figure 5 (B)-(E), where the magnitude of the heatmap represents the correlation between EEG channels (rows) and visual sub-features (columns). Higher values denote a greater selectivity of the channel signal for a specific sub-feature.

From Figure 5 (A) we have two observations. **(1) Early brain selectivity**: the strongest brain activation selectivity occur within 0–400 ms and then gradually attenuate. Specifically, at 0–100 ms the per-row mapping weights span approximately 0.40–0.54. The brain activity exhibits a consistent spatiotemporal progression, shifting from the posterior occipital to the parietal cortex, a pattern that is consistent with the sequential engagement of early- to mid-level visual areas (Ungerleider, 1982). **(2) Time-varying channel selectivity.** The same channel exhibits different sub-feature preferences across time. For instance, the *Oz* channel attends more to the 5-th sub-feature (and less to the 8-th in the earliest window, but by the third window this preference reverses, suggesting that brain channels process visual signals with time-varying channel selectivity.

**Visualization of Brain-Visual Alignment Patterns.** As shown in Figure 6, we visualize the fine-grained channel-to-visual mapping matrices from CCFP of all subjects. The visualizations reveal several consistent correlation patterns between EEG channels and visual sub-features: (1) Across the majority of subjects, the *Oz* channel is strongly associated with the 13-th sub-feature, which is the background-linked component dominated by low-frequency power (see 3.6). (2) Channels *P7* and *P8*, located symmetrically over the parietal–occipital cortex, tend to associate with the 0-th sub-feature (foreground-related; see 3.6), despite generally modest activations across most P-region channels. This pattern is evident in the heatmaps of `subject 1` to `subject 4`, with stronger effects at *P8* for `subject 6` and at *P7* for `subject 10`, potentially reflecting hemispheric differ-

ences or stimulus spatial configuration. (3) Most subjects show a clear association between channel *O1/O2* and the 2-nd sub-feature, which is foreground-related and predominantly high-frequency. In addition, *PO3/PO4* in most subjects are linked to background-related features. These observations offer interpretability for BRENA's improvements, indicating that it captures functional specialization through meaningful local mappings between brain regions and visual patterns. By incorporating region-specific selectivity, BRENA grounds its alignment in human perceptual mechanisms, thereby strengthening its overall effectiveness.

### 3.6 QUALITATIVE VISUALIZATION

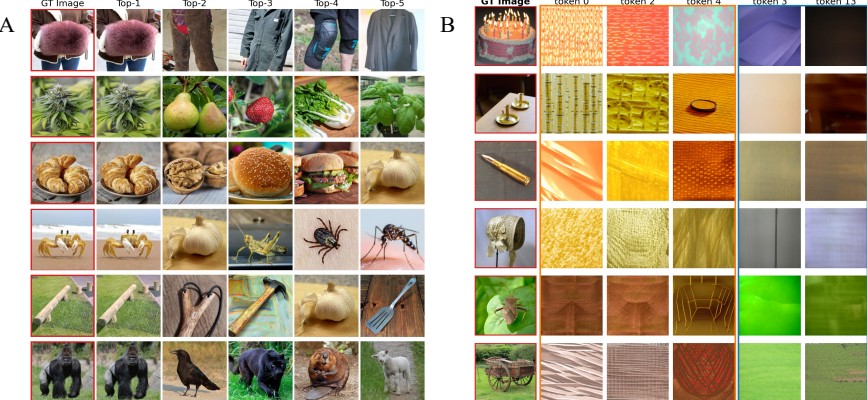

Figure 7: Visualization of retrieval performance and target visual sub-feature visualization. (A) Top-5 retrieval examples on sub-01. (B) Target visual sub-feature visualization of foreground (orange box) and background (blue box). The target sub-feature indices are labeled at the top of the Figure.

**Visualization of Retrieval Results.** As shown in Figure 7 (A), our method produces features that precisely match the target image while also retrieving visually similar images using cues such as color, contour, and texture. Our approach also demonstrates strong intra-class consistency across broad categories (e.g., animals, insects, foods, clothing). Notably, the top-similarity retrievals consistently remain within the query's coarse category (e.g., a "gorilla" query retrieves other animals). The semantic consistency also extends beyond category, with retrievals also reflecting structural geometry (a "crab" retrieves a similar insect), material and orientation (a "balance beam" retrieves slingshots and hammers), and color (a "black orangutan" retrieves a black crow and a black panther). These observations indicate that our approach maintains semantic consistency across multiple levels, from category membership to fine-grained attributes.

**Visualization of Visual Sub-features.** To probe the semantics of different target sub-features, we visualize the 0th, 2nd, 4th, 3rd, and 13th sub-features across various images in Figure 7 (B). These sub-features broadly fall into two categories: *foreground-related* (0, 2, 4) and *background-related* (3, 13). Some sub-features consistently express **high-frequency texture** (dense stripes/ridges, fine repetitive patterns, sharp micro-edges), some capture **low-frequency background** (dark fields, defocus, broad color washes). Crucially, these preferences persist across different images, indicating that each sub-feature functions as a low-level semantic detector. The results show why using visual sub-features outperforms a single global embedding: sub-features provide decomposed, complementary evidence that the alignment can exploit, avoiding over-pooled semantics.

## 4 CONCLUSION

We present BRENA, a hierarchical brain–vision alignment framework that integrates an adaptive *local* module with a *global* alignment module. The local alignment module aligns EEG channel embeddings to semantically resampled visual sub-features guided by adaptive generated perceptual weights, enabling more effective use of EEG for visual decoding. Extensive experiments demonstrate the effectiveness of our method, boosting Top-1 accuracy by 7.3% over SOTA on the 200-way retrieval task in intra-subject setting.

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

# APPENDIX

## A  RELATED WORK

### A.1  BRAIN DECODING

Early progress in visual decoding was largely driven by fMRI-based approaches (Smith, 2004), benefiting from their high spatial fidelity and relative robustness in capturing neural responses. Mind-Eye (Scotti et al., 2023) introduced a two-branch paradigm with diffusion priors that became a foundation for later work. Subsequent work (Chen et al., 2023a) advanced this line by adopting masked brain modeling to enhance fMRI representations. Building on this foundation, later studies improved data efficiency and transferability (Scotti et al., 2024), optimized system latency (Benchetrit et al., 2023), and extended decoding from static images to dynamic video (Chen et al., 2023b; Gong et al., 2024; Lu et al., 2025). The release of the THINGS-EEG dataset (Gifford et al., 2022; Song et al., 2023) presents experimental evidence for the feasibility of EEG-based visual decoding. ATM (Li et al., 2024a) designs an encoder with channel attention and spatiotemporal convolutions, achieving promising performance on retrieval and reconstruction. Building on this line, early efforts concentrated on alignment procedures and on strengthening brain encoders (Wei et al., 2024; Li et al., 2024b). More recent works have shifted toward the target side by redefining the alignment objective, broadening it beyond the original image. (Zhang et al., 2025) augments supervision with auxiliary depth and text representations, while (Liu et al., 2025) injects segmentation-style structural cues. UBP (Wu et al., 2025) reveals discrepancies between visual models and human perception, addressing them with an uncertainty-aware blur prior that boosts retrieval performance. However, most prior works rely on global alignment for EEG–visual bridging, overlooking the functional specialization of perception.

### A.2  CROSS-MODAL ALIGNMENT

Cross-modal alignment is typically achieved by constructing a shared embedding space where paired modalities are mapped to nearby representations. A seminal example is CLIP (Radford et al., 2021), which employs a ViT image encoder and a transformer text encoder to project image patches and text into a joint embedding space. Trained with an InfoNCE-style contrastive loss (Oord et al., 2018), CLIP demonstrates strong zero-shot classification, image–text retrieval, and even serves as a conditioning backbone for diffusion models. This paradigm has since become a standard foundation for multimodal learning, inspiring extensions across diverse domains beyond vision–language tasks. In brain decoding, such contrastive alignment approach has been widely adopted (Li et al., 2024a; Scotti et al., 2024). However, the substantial heterogeneity between brain signals and visual data poses a central challenge on how to design a reasonable alignment mechanism that effectively capture the cognitive patterns of neural activity. To this end, BRENA introduces a hierarchical neural alignment framework with multi-level semantics explored.

## B  PREPROCESSING DETAILS

We used the publicly available dataset introduced by Gifford et al. (2022), which contains EEG recordings from ten participants under a rapid serial visual presentation (RSVP) paradigm. Each stimulus was displayed for 100 ms, followed by a 100 ms blank interval. Raw EEG signals were recorded from 63 channels at a sampling rate of 1000 Hz and filtered within the [0.1, 100] Hz band.

For preprocessing, EEG data were segmented into trials spanning 0–1000 ms relative to stimulus onset. To reduce computational load while retaining temporal information, the signals were down-sampled to 250 Hz. Only visual-related channels were preserved in our experiment. Multivariate noise normalization was applied using training data, following the approach of Guggenmos et al. (2018), to mitigate channel-wise variability and improve generalization. To enhance the signal-to-noise ratio, we averaged repeated EEG trials corresponding to the same image stimulus.

## C  IMPLEMENTATION DETAILS

Our method was implemented in Python 3.10.14 with PyTorch 2.8.0+cu128 and cuDNN 9.1.2. All experiments were conducted on a server equipped with an AMD EPYC 7542 CPU (128 cores), a single NVIDIA A800 GPU, and 503 GB of RAM. Models were trained for 50 epochs with a batch size of 1024. We used the AdamW optimizer with weight decay. The learning rate was set to $1 \times 10^{-4}$ for the intra-subject setting and $1 \times 10^{-5}$ for the inter-subject setting. We applied early stopping based on validation performance to prevent overfitting.

## D  ADDITIONAL QUANTITATIVE RESULTS

In this section, we provide more detailed quantitative results to complement the main text. While Sections 3.2, 3.3, and 3.4 report the overall comparisons and key ablations, here we present the detailed per-subject retrieval accuracies, results across different sub-feature and channel combinations.

Table 2: Top-1 and Top-5 accuracy (%) for 200-way zero-shot retrieval with different models.

| Method | Subject 1 Top-1 | Top-5 | Subject 2 Top-1 | Top-5 | Subject 3 Top-1 | Top-5 | Subject 4 Top-1 | Top-5 | Subject 5 Top-1 | Top-5 | Subject 6 Top-1 | Top-5 | Subject 7 Top-1 | Top-5 | Subject 8 Top-1 | Top-5 | Subject 9 Top-1 | Top-5 | Subject 10 Top-1 | Top-5 | Avg Top-1 | Top-5 |
|---|---|---|---|---|---|---|---|---|---|---|---|---|---|---|---|---|---|---|---|---|---|---|
| **Intra-subject** | | | | | | | | | | | | | | | | | | | | | | |
| vit b 16 | 30.0 | 54.5 | 20.0 | 53.0 | 23.5 | 54.5 | 31.5 | 58.0 | 21.5 | 41.5 | 32.5 | 58.0 | 32.0 | 64.0 | 32.0 | 65.5 | 25.5 | 46.5 | 34.0 | 56.5 | 28.25 | 55.25 |
| vit b 32 | 35.5 | 67.0 | 24.5 | 58.0 | 33.5 | 63.5 | 33.5 | 64.0 | 25.5 | 52.0 | 35.0 | 67.0 | 37.5 | 67.5 | 35.0 | 76.5 | 27.0 | 53.5 | 35.5 | 73.5 | 32.6 | 64.25 |
| vit l 14 | 20.5 | 48.0 | 14.0 | 35.5 | 19.5 | 44.5 | 25.5 | 55.5 | 14.5 | 32.5 | 17.5 | 48.0 | 20.5 | 50.5 | 22.0 | 56.5 | 15.5 | 35.0 | 20.5 | 58.5 | 19.0 | 46.45 |
| dino | 45.0 | 85.0 | 27.5 | 63.5 | 48.0 | 76.5 | 48.0 | 78.5 | 28.0 | 56.5 | 51.5 | 78.5 | 42.5 | 74.5 | 57.0 | 84.5 | 28.0 | 61.0 | 42.0 | 81.0 | 41.45 | 73.45 |
| dinov2 | 37.0 | 72.5 | 31.5 | 64.5 | 42.0 | 76.5 | 44.5 | 72.0 | 32.0 | 57.0 | 39.5 | 67.5 | 47.0 | 78.5 | 54.0 | 78.5 | 33.5 | 63.5 | 52.0 | 79.0 | 40.4 | 70.0 |
| blip | 32.0 | 58.5 | 30.0 | 54.0 | 35.5 | 68.0 | 37.0 | 69.0 | 30.0 | 57.5 | 35.5 | 65.5 | 32.0 | 60.5 | 44.5 | 73.5 | 34.5 | 60.0 | 42.0 | 69.5 | 34.3 | 63.85 |
| open-flamingo | 33.0 | 62.0 | 29.5 | 62.0 | 31.0 | 62.0 | 33.0 | 68.0 | 24.0 | 51.5 | 28.5 | 60.0 | 33.5 | 67.0 | 45.5 | 74.0 | 31.5 | 58.5 | 38.5 | 69.0 | 32.6 | 63.4 |
| **Inter-subject** | | | | | | | | | | | | | | | | | | | | | | |
| vit b 16 | 10.5 | 27.0 | 11.0 | 29.5 | 8.0 | 25.5 | 11.5 | 33.5 | 12.0 | 25.0 | 11.0 | 22.0 | 13.5 | 32.0 | 10.0 | 25.5 | 12.0 | 38.5 | 15.5 | 37.5 | 11.5 | 29.8 |
| vit b 32 | 13.0 | 34.5 | 11.5 | 33.5 | 9.5 | 27.5 | 15.0 | 34.5 | 10.0 | 31.5 | 10.5 | 29.5 | 13.0 | 30.5 | 7.5 | 30.5 | 11.0 | 34.0 | 14.0 | 33.5 | 11.6 | 32.6 |
| vit l 14 | 9.5 | 27.0 | 7.5 | 17.0 | 7.5 | 24.0 | 10.0 | 28.0 | 8.5 | 16.0 | 8.5 | 24.5 | 9.5 | 30.5 | 5.0 | 25.5 | 9.0 | 17.0 | 8.5 | 29.0 | 8.35 | 23.85 |
| dino | 27.0 | 47.0 | 17.5 | 42.0 | 15.0 | 38.5 | 19.0 | 46.5 | 19.0 | 39.5 | 14.5 | 35.5 | 14.0 | 50.0 | 17.0 | 55.5 | 24.5 | 46.5 | 29.0 | 57.5 | 19.6 | 45.15 |
| dinov2 | 18.5 | 47.0 | 19.0 | 42.5 | 15.0 | 42.5 | 18.5 | 41.5 | 8.5 | 36.5 | 14.5 | 29.0 | 15.0 | 42.0 | 16.0 | 40.5 | 21.0 | 47.5 | 18.0 | 53.5 | 16.4 | 42.8 |
| blip | 8.0 | 27.0 | 8.5 | 29.5 | 6.5 | 23.5 | 8.5 | 24.5 | 9.0 | 20.5 | 8.0 | 20.5 | 9.0 | 25.0 | 7.5 | 30.0 | 9.5 | 32.5 | 9.5 | 31.5 | 8.4 | 26.35 |
| open-flamingo | 11.0 | 30.0 | 11.5 | 36.5 | 10.0 | 19.5 | 13.5 | 34.0 | 7.5 | 26.5 | 12.0 | 18.0 | 12.5 | 20.5 | 16.0 | 26.5 | 14.0 | 34.5 | 20.0 | 34.5 | 12.9 | 28.7 |

Table 3: Top-1 and Top-5 accuracy (%) for 200-way zero-shot retrieval using different token combinations.

| Method | Subject 1 Top-1 | Top-5 | Subject 2 Top-1 | Top-5 | Subject 3 Top-1 | Top-5 | Subject 4 Top-1 | Top-5 | Subject 5 Top-1 | Top-5 | Subject 6 Top-1 | Top-5 | Subject 7 Top-1 | Top-5 | Subject 8 Top-1 | Top-5 | Subject 9 Top-1 | Top-5 | Subject 10 Top-1 | Top-5 | Avg Top-1 | Top-5 |
|---|---|---|---|---|---|---|---|---|---|---|---|---|---|---|---|---|---|---|---|---|---|---|
| **Intra-subject** | | | | | | | | | | | | | | | | | | | | | | |
| f | 52 | 82.5 | 43.5 | 75 | 50.5 | 77 | 43.0 | 75.5 | 33.0 | 62 | 51.0 | 74.5 | 52.5 | 81.5 | 55.5 | 83.0 | 36.0 | 71.5 | 57.0 | 85.5 | 47.4 | 76.8 |
| b | 35.5 | 66 | 31.0 | 57.5 | 28.5 | 61.5 | 29.5 | 59 | 21.0 | 50.5 | 33.5 | 60.5 | 27.5 | 56.0 | 34.5 | 65.5 | 24.0 | 52.5 | 33.5 | 70.5 | 29.9 | 59.9 |
| oth | 33.5 | 64.5 | 29.0 | 59 | 32.0 | 63 | 30.0 | 60.5 | 25.0 | 49.5 | 31.5 | 65.5 | 31.0 | 61.5 | 30.5 | 61 | 41.0 | 70.5 | | | 31.9 | 61.3 |
| f+b | 56.5 | 84.5 | 51.0 | 82.5 | 55.0 | 86 | 45.5 | 77 | 40.5 | 67.5 | 57.5 | 79.5 | 54.5 | 85.5 | 60.0 | 84.5 | 45.5 | 80.5 | 60.0 | 86 | 52.6 | 81.4 |
| f+oth | 56.5 | 84.0 | 50.0 | 78.0 | 54.0 | 81.0 | 46.5 | 79.5 | 37.5 | 68.0 | 53.0 | 82.5 | 57.5 | 83.0 | 58.5 | 84.5 | 43.5 | 80.5 | 60.5 | 85.0 | 51.8 | 80.3 |
| b+oth | 42.0 | 75.5 | 37.5 | 64.5 | 34.5 | 65.5 | 35.5 | 67 | 27.0 | 54.5 | 33.0 | 68 | 35.0 | 62.5 | 43.0 | 76.5 | 31.0 | 61 | 44.5 | 78.5 | 36.3 | 67.4 |
| all | 61.0 | 87.0 | 56.0 | 81.5 | 56.5 | 87.5 | 57.0 | 84.5 | 46.5 | 74.5 | 56.5 | 87.0 | 58.5 | 86.0 | 75.0 | 91.5 | 50.0 | 80.0 | 65.0 | 92.0 | **58.2** | **85.1** |
| **Inter-subject** | | | | | | | | | | | | | | | | | | | | | | |
| f | 13.0 | 35.5 | 12.5 | 33.5 | 9.0 | 21.0 | 8.0 | 24.0 | 8.0 | 26.5 | 4.5 | 17.5 | 9.0 | 28.5 | 10.5 | 26.5 | 12.0 | 29.5 | 14.0 | 32.0 | 10.1 | 27.5 |
| b | 8.5 | 26.5 | 10.5 | 29.5 | 4.5 | 19.0 | 4.5 | 13.5 | 5.0 | 18.5 | 7.0 | 18.5 | 6.0 | 20.0 | 7.0 | 20.0 | 8.5 | 27.5 | 11.0 | 30.5 | 7.2 | 22.4 |
| oth | 7.5 | 26.0 | 8.0 | 29.0 | 3.5 | 17.5 | 7.5 | 19.0 | 11.5 | 24.5 | 4.5 | 16.5 | 8.0 | 31.0 | 7.5 | 20.0 | 10.5 | 32.0 | 11.0 | 31.5 | 7.9 | 24.0 |
| f+b | 12.5 | 31.0 | 13.0 | 32.5 | 12.0 | 29.0 | 9.5 | 29.0 | 6.0 | 24.5 | 9.5 | 23.0 | 10.0 | 32.0 | 10.0 | 26.5 | 13.0 | 33.5 | 13.0 | 37.0 | 10.6 | 29.8 |
| f+oth | 14.5 | 35.5 | 11.0 | 32.0 | 6.5 | 29.0 | 10.0 | 33.5 | 9.5 | 23.5 | 6.5 | 29.5 | 9.5 | 31.5 | 10.0 | 30.0 | 11.5 | 32.0 | 16.0 | 34.0 | 10.3 | 29.8 |
| b+oth | 8.0 | 20.0 | 8.5 | 28.5 | 4.0 | 21.0 | 9.0 | 22.5 | 6.0 | 22.0 | 5.5 | 22.0 | 7.0 | 24.5 | 9.0 | 26.0 | 12.0 | 32.0 | 9.5 | 26.0 | 7.7 | 23.5 |
| all | 15.0 | 37.5 | 18.5 | 47.0 | 11.0 | 31.5 | 18.5 | 39.5 | 13.5 | 35.0 | 9.5 | 27.5 | 13.5 | 36.0 | 15.0 | 40.0 | 18.0 | 42.0 | 24.5 | 50.0 | 15.7 | **38.7** |

Table 4: Top-1 and Top-5 accuracy (%) for 200-way zero-shot retrieval with different channel combinations.

| Method | Subject 1 | | Subject 2 | | Subject 3 | | Subject 4 | | Subject 5 | | Subject 6 | | Subject 7 | | Subject 8 | | Subject 9 | | Subject 10 | | Avg | |
|---|---|---|---|---|---|---|---|---|---|---|---|---|---|---|---|---|---|---|---|---|---|---|
| | Top-1 | Top-5 | Top-1 | Top-5 | Top-1 | Top-5 | Top-1 | Top-5 | Top-1 | Top-5 | Top-1 | Top-5 | Top-1 | Top-5 | Top-1 | Top-5 | Top-1 | Top-5 | Top-1 | Top-5 | Top-1 | Top-5 |
| **Intra-subject** | | | | | | | | | | | | | | | | | | | | | | |
| o | 32.0 | 66.5 | 26.5 | 56.5 | 32.5 | 65.0 | 28.5 | 60.0 | 21.5 | 48.5 | 29.0 | 61.5 | 31.0 | 63.0 | 38.5 | 69.0 | 34.5 | 60.5 | 36.5 | 71.0 | 31.1 | 62.2 |
| p | 15.0 | 39.0 | 23.5 | 50.5 | 24.0 | 50.0 | 28.0 | 56.5 | 18.0 | 46.5 | 24.0 | 52.5 | 20.5 | 43.0 | 46.0 | 80.0 | 18.5 | 40.5 | 29.0 | 61.0 | 24.7 | 52.0 |
| po | 45.0 | 73.0 | 44.0 | 71.5 | 45.0 | 76.5 | 38.5 | 71.5 | 31.0 | 60.5 | 40.5 | 69.5 | 37.0 | 65.5 | 49.0 | 80.5 | 29.0 | 59.0 | 44.0 | 73.0 | 40.3 | 70.1 |
| o+p | 48.5 | 76.5 | 41.0 | 73.5 | 46.0 | 79.0 | 48.0 | 73.5 | 33.5 | 63.5 | 51.0 | 83.0 | 45.5 | 71.5 | 67.5 | 91.0 | 41.5 | 71.5 | 50.0 | 82.5 | 47.3 | 76.6 |
| p+po | 45.0 | 78.5 | 40.5 | 71.5 | 48.0 | 78.5 | 43.0 | 78.0 | 37.0 | 65.5 | 41.0 | 73.0 | 38.0 | 70.5 | 63.0 | 88.0 | 29.0 | 61.5 | 51.0 | 79.0 | 43.6 | 74.4 |
| o+po | 58.0 | 90.0 | 53.0 | 81.5 | 54.0 | 84.0 | 53.5 | 85.0 | 41.0 | 71.5 | 58.5 | 86.5 | 57.0 | 86.0 | 63.0 | 87.5 | 51.0 | 80.5 | 63.5 | 88.5 | 55.3 | 84.1 |
| o+p+po | 61.0 | 87.0 | 56.0 | 81.5 | 56.5 | 87.5 | 57.0 | 84.5 | 46.5 | 74.5 | 56.5 | 87.0 | 58.5 | 86.0 | 75.0 | 91.5 | 50.0 | 80.0 | 65.0 | 92.0 | **58.2** | **85.1** |
| **Inter-subject** | | | | | | | | | | | | | | | | | | | | | | |
| o | 11.5 | 33.0 | 7.5 | 22.0 | 5.0 | 15.5 | 7.0 | 22.5 | 4.5 | 22.0 | 3.0 | 13.5 | 5.5 | 16.0 | 9.0 | 18.5 | 8.0 | 23.0 | 5.0 | 22.0 | 6.6 | 20.8 |
| p | 2.0 | 13.5 | 6.5 | 22.0 | 7.5 | 20.5 | 7.0 | 22.5 | 5.0 | 19.0 | 5.5 | 17.0 | 7.0 | 15.5 | 6.0 | 23.0 | 3.5 | 18.0 | 3.0 | 21.5 | 5.4 | 19.3 |
| po | 7.0 | 29.0 | 9.5 | 22.5 | 5.0 | 20.5 | 8.5 | 21.0 | 4.5 | 14.5 | 4.5 | 14.5 | 7.0 | 19.0 | 6.0 | 22.5 | 3.5 | 19.5 | 9.5 | 25.0 | 6.5 | 20.8 |
| o+p | 9.0 | 32.5 | 15.0 | 36.0 | 11.5 | 25.0 | 15.0 | 33.5 | 8.0 | 26.5 | 10.0 | 27.5 | 10.0 | 25.5 | 13.0 | 32.5 | 16.0 | 30.5 | 15.0 | 31.5 | 12.3 | 31.2 |
| p+po | 9.0 | 25.0 | 8.5 | 24.0 | 9.5 | 27.5 | 11.5 | 28.0 | 7.5 | 22.5 | 4.5 | 18.5 | 7.0 | 27.0 | 11.0 | 30.5 | 7.5 | 25.5 | 12.5 | 29.5 | 9.2 | 25.8 |
| o+po | 13.5 | 35.5 | 12.5 | 35.0 | 11.0 | 26.0 | 10.5 | 28.0 | 9.5 | 28.5 | 9.0 | 23.0 | 9.0 | 33.5 | 12.5 | 36.5 | 13.0 | 30.5 | 13.0 | 34.5 | 11.4 | 31.1 |
| o+p+po | 15.0 | 37.5 | 18.5 | 47.0 | 11.0 | 31.5 | 18.5 | 39.5 | 13.5 | 35.0 | 9.5 | 27.5 | 13.5 | 36.0 | 15.0 | 40.0 | 18.0 | 42.0 | 24.5 | 50.0 | **15.7** | **38.7** |

# E  THE USE OF LARGE LANGUAGE MODELS

Large language models (LLMs) were used to aid the writing of this work. LLMs helped improve clarity and grammar of the manuscript. LLMs were not used for literature retrieval, related work discovery, or research ideation.

