# OpenReview forum: "BRENA: Brain-inspired Hierarchical Neural Alignment Framework for Visual Decoding from EEG Signals"
_ICLR.cc/2026/Conference — ICLR 2026 Conference Withdrawn Submission_

### Official Review · Reviewer_R9aJ · 2025-10-26

**Soundness:** 2
**Presentation:** 2
**Contribution:** 2
**Rating:** 2
**Confidence:** 4

**Summary:**

Decoding visual content from brain signals means trying to figure out what a person is seeing based on their brain activity. This has been a research focus for decades. This paper introduces a new framework called BRENA (BRain-inspired hiErarchical Neural Alignment), which aims to improve brain decoding by aligning brain activity and visual information at both the region level and the overall level. The motivation behind this work is clearly explained.

**Strengths:**

Clearly presented；
Well defined problem;

**Weaknesses:**

Please refer to questions.

**Questions:**

While the method is clearly described, it doesn’t offer much technical novelty. Similar ideas have already been explored in many past studies published at major conferences like ICLR, CVPR, ICCV, or in relevant journals. Clearly, the authors miss a large portion of literature.

The paper doesn’t clearly explain how the baseline methods for comparison were chosen, or why specific design choices—like the type of visual encoder used—were made. More experiments like ablation studies are needed to test how different visual encoders or key settings like the number of sub-features affect performance.

The results should also be more thoroughly analyzed. For example, although BRENA performs better than other methods when tested across different persons (inter-subject), the actual accuracy is still quite low regarding top-1 accuracy, compared to 58.2% when testing within the same person (intra-subject). This shows that the model still struggles with large differences between people’s brain structures and functions, which is a big unsolved problem in the field. Again, there are numerous studies on improving inter-subject generalization.

The model was only tested on zero-shot retrieval using one dataset and under one specific setup. It’s unclear how well the method would work on other important tasks like image generation from EEG or with different kinds of visual inputs (like videos, faces, or scenes). Some extra results are included in the appendix, but they add little value.

The paper also discusses some neuroscience-related insights (for example, the Oz channel being linked to background processing), which come from visualizing the model's learned weights. These findings are interesting and seem to match existing knowledge, but they only show correlation, not cause and effect. Therefore, the conclusion is not rigorous. The model itself isn’t naturally interpretable, say, understanding its behavior requires extra visual analysis steps.

---

### Official Review · Reviewer_Rrky · 2025-10-29

**Soundness:** 2
**Presentation:** 3
**Contribution:** 2
**Rating:** 4
**Confidence:** 4

**Summary:**

This paper proposes the BRENA method, to align human brain EEG signals and visual features from perceived static images.
Unlike previous approaches which contrastively align all EEG signals to all visual features globally, they align sub-features from images with EEG channels selectively, combined with a CLIP loss for aligning the brain and the image features globally. This improves image-retrieval performance (top-1 and top-5 accuracy) across a number of previous baselines on the THINGS-EEG dataset. The method also offers some level of interpretability for the mapping of specific EEG channels to specific visual features, and on how this mapping changes over time after the presentation of an image stimulus.

**Strengths:**

- The paper is clearly written, figures and methods are described in detail.
- The motivation and contrast with previous approaches is clear and grounded in insights from experimental neuroscience.
- The claims are well-supported by the figures and the clear performance improvements on baselines on the THINGS-EEG across all subjects.

**Weaknesses:**

- The main results and ablations are missing variability estimates / statistical testing. Since some subjects or ablations show relatively small differences in scores (< 1 %), it is not obvious whether this difference is statistically significant.
- The use of a single EEG / Image dataset makes it difficult to evaluate the generalisability of this approach. There are other publicly available EEG/Image datasets (e.g. Grootswaggers et al) and a complementary successful application of BRENA to another dataset would reinforce its soundness and contribution.
- The baseline reproduction is missing details, in particular on whether the authors had to slightly deviate from the original baseline implementations.

**Questions:**

- Fig 7 only shows perfect retrieval results, showing failure cases with an additional discussion of these would reinforce the strength of these positive results.
- It is not clear what $f$ and $\psi$ are in the CCFP module description. Also, the CCFP could be interpreted as if all EEG channels are actually pooled at this level (dimensionality map `Cxm -> Kxn`). While the author's approach obviously improves performance against baselines, this apparent pooling step makes it unclear whether the contribution is about per-channel per-subfeature alignment versus a localised alignment of a mix of EEG features.
- The nature of the local projector $\mathcal{M}$ is not given, and it could be clarified weather its weights are shared across all N semantic embeddings.


Minor comments:
- The fact that the paper focuses on the task of EEG to image decoding could be clarified in the abstract and earlier in the introduction.
- Intro suggests Scotti and Benchetrit studies are focusing on EEG, whereas they respectively deal with fMRI and MEG (this is correctly stated in the Appendix - Related works section).
- Aligning the Y-axis of panels A/B from Figure 3 would clarify the loss in performance between the settings ‘single-subject’ and ‘train on all subjects but subj_i and test on subj_i’.
- More details could be provided on the Gifford dataset (e.g. number of repeated presentations for train / test splits)
- The values of lambda / temperature could be explicitly specified.

---

### Official Review · Reviewer_heZE · 2025-10-29

**Soundness:** 3
**Presentation:** 2
**Contribution:** 2
**Rating:** 4
**Confidence:** 4

**Summary:**

The paper addresses brain decoding and presents a  BRain-inspired hiErarchical Neural Alignment (BRENA) framework by simultaneously aligning both region-level and global brain representations with visual embeddings.  BRENA proposes an Adaptive Local Neural Alignment Module to explore correspondence between brain channel features and visual semantic units with a goal of  better exploitation of brain
signals by modelling region-specific feature selectivity. Furthermore, a global neural alignment module is explored to achieve hierarchical
brain-visual alignment, capturing complementary region-level and global neural patterns.

**Strengths:**

The presented hierarchical brain–vision alignment framework's main contribution is integration of an adaptive local module with a global alignment module. This local alignment module aligns EEG channel embeddings to semantically resampled visual sub-features. The sub-features are guided by adaptive generated perceptual weights, enabling more effective use of EEG for visual decoding. The idea of the paper of using global and local is not fully original given that the adaptation and alignment are commonly used when multilayered systems are facilitated but not always considered as local and global modules per sig. Paper is clearly written, results have some significance on the rather limited dataset. .

**Weaknesses:**

The authors fail to provide a balanced discussion on strengths and weaknesses of the proposed method. I would also encourage authors to provide a more in-depth motivation of 'brain-insipred' aspect of the work - what particular studies did you build your model on.

**Questions:**

Regarding inter-subject performance: an increase of   5.3 points over the strongest prior baseline is achieved and  Top-1 accuracy of 15.7%. It Based on this, it is claimed that BRENA learns shared structure across subjects more effectively. But is it really effective or just better that the baseline?

The improvements are also shown on intra-subject performance in relation to the baseline but Top-1 accuracy is still a big challenge. What is the biggest reason for  still rather mediocre performance in the proposed model?

---

### Official Review · Reviewer_3htD · 2025-11-05

**Soundness:** 1
**Presentation:** 2
**Contribution:** 1
**Rating:** 2
**Confidence:** 4

**Summary:**

The paper presents a deep learning pipeline for brain-to-image decoding from EEG data. The proposed method relies on alignment at both “local” and “global” levels, i.e., a neural encoder is trained to align EEG representations to both visual “sub-features”, and to a global visual representation of the image. At inference time, retrieval is performed on the concatenated representations. Experiments on a dataset comprising 10 subjects suggest the proposed approach outperforms existing baselines in intra- and inter-subject settings. Finally, ablation studies on the main components of the models and visualizations of retrieved images are shown.

**Strengths:**

Originality:
* The decomposition of brain-to-image alignment into multiple parallel image representations as a way to improve decoding performance appears novel.

Quality:
* The comparison in image decoding performance obtained for different image backbones (Figure 3) is informative.

Significance:
* The proposed method appears to outperform existing methods systematically on a retrieval task. If these results hold on different datasets and on an image reconstruction task, this approach is likely to be beneficial in standard brain decoding pipelines in general and not only for image decoding.

**Weaknesses:**

1. I found the methodology difficult to understand as many technical aspects are described somewhat fuzzily. For instance, it is unclear what kind of architectures and numbers of parameters the channel embedder, cross-channel fusion projector, and encoder-resampler have, and what the final parameter count of the proposed model is. See Q1-4.

2. To the best of my understanding, the main claim of the paper, i.e. that using region-specific features helps improve visual decoding performance, appears to be contradicted by the design choice of the cross-channel fusion projector. Given this projector mixes information across channels, the alignment with visual features (happening inside the local alignment module) can no longer be considered to be local. See Q5.

3. Claims on the importance of different design choices are weakly supported given the lack of variability estimates in the results. For instance, in Figure 4C, does k have a statistically significant impact on performance (especially on top-5 accuracy)? Similarly, in Figure 4D, half of the subjects don’t seem to benefit from the local alignment module, despite the text claiming that this module "consistently improves Top-1 accuracy across all ten subjects”. Proper statistical analysis could reveal that these design choices have a limited effect on performance.

4. The paper only presents results on a single dataset, which weakens the validation of the approach and results. There are similar public datasets that may be of interest, e.g. [1, 2].

[1] Grootswagers, Tijl, et al. "Human EEG recordings for 1,854 concepts presented in rapid serial visual presentation streams." Scientific Data 9.1 (2022): 3.
[2] Xu, Jonathan, et al. "Alljoined-1.6 M: A Million-Trial EEG-Image Dataset for Evaluating Affordable Brain-Computer Interfaces." arXiv preprint arXiv:2508.18571 (2025).

5. The paper only presents results on an image retrieval task, whereas the field of brain-to-image decoding has been producing results on the image reconstruction task for a few years, e.g. in Benchetrit et al. (2023), Li et al. (2024) and Zhang et al. (2025). Showing that the proposed approach also improves image reconstruction performance is important for its validation.

6. Care must be taken when directly interpreting the learned weights of linear backward projections (see Figures 5 and 6, where projection weights from EEG channels to visual sub-features are visualized). See [3] for a description of this problem. The analyses of Figures 5 and 6 should be double checked to ensure similar conclusions hold when looking at the equivalent forward models.

[3] Haufe, Stefan, et al. "On the interpretation of weight vectors of linear models in multivariate neuroimaging." Neuroimage 87 (2014): 96-110.

**Questions:**

1. What are $\phi_f$ and $\phi_t$ at p.4, l.169? Also, I find it surprising that the temporal and spectral representations are directly summed to the input raw signal. Is this actually better than a simple concatenation operation? References or an ablation study on this design choice would be informative.
2. What is the architecture and number of parameters of the “Cross-Channel Fusion Projector”? There are two mappings ($f$ and $\psi$), but the text doesn’t specify what they are. My understanding is that the CCFP is carrying out a sort of aggregation of the different channels. Also, I don’t see which part of Figure 1 the text refers to: “As shown in Figure 1, by visualizing the weight matrix of the fusion mapping layer, we can identify EEG channels that contribute most to specific sub-visual features and compare the relative strength of channel-feature couplings across brain regions.”
3. What is M in Equation 3, i.e. its architecture and number of parameters?
4. Some clarification questions about the “Encoder/Resampler”: First, just to better understand the scope of this work, where does the pretrained visual encoder checkpoint come from? Second, I don’t understand whether the visual resampler is part of the original encoder, some follow-up external work, or whether it was actually learned from scratch as part of the alignment process, as I don’t see the term “resampler” mentioned in the reference (Ye et al., 2023). Ultimately, what kind of features are returned by this resampler? If trained from scratch, why not reduce the number of cross-attention queries to k?
5. Local alignment: I understand that each visual sub-feature is then aligned with the output of a linear projection (which by the way is not defined, e.g. architecture and number of parameters). However, at this point, the neural data has been aggregated in the spatial dimension by the CCFP and potentially the domain adapter. If so, can these actually correspond to specific brain regions (p.4, line 212)?
6. What is the unit on Fig 5A? Also, do I understand correctly that this is the time-locked amplitude averaged over all examples of the training set for Subject 1? Of note, the text, especially on the colorbars, is very small and difficult to read.

---

### Note · Authors · 2025-11-13

**Comment:**

Thank all the reviewers for their valuable comments.

**Withdrawal Confirmation:**

I have read and agree with the venue's withdrawal policy on behalf of myself and my co-authors.